# Modulating the hierarchical fibrous assembly of Au nanoparticles with atomic precision

Qi Li[1], Jake C. Russell [2], Tian-Yi Luo[3], Xavier Roy[2], Nathaniel L. Rosi[3], Yan Zhu [4] & Rongchao Jin[1]

The ability to modulate nanoparticle (NP) assemblies with atomic precision is still lacking, which hinders us from creating hierarchical NP organizations with desired properties. In this work, a hierarchical fibrous (1D to 3D) assembly of Au NPs (21-gold atom, $Au_{21}$) is realized and further modulated with atomic precision via site-specific tailoring of the surface hook (composed of four phenyl-containing ligands with a counteranion). Interestingly, tailoring of the associated counterion significantly changes the electrical transport properties of the NP-assembled solids by two orders of magnitude due to the altered configuration of the interacting $\pi$–$\pi$ pairs of the surface hooks. Overall, our success in atomic-level modulation of the hierarchical NP assembly directly evidences how the NP ligands and associated counterions can function to guide the 1D, 2D, and 3D hierarchical self-assembly of NPs in a delicate manner. This work expands nanochemists' skills in rationally programming the hierarchical NP assemblies with controllable structures and properties.

[1] Department of Chemistry, Carnegie Mellon University, Pittsburgh, PA 15213, USA. [2] Department of Chemistry, Columbia University, New York, NY 10027, USA. [3] Department of Chemistry, University of Pittsburgh, Pittsburgh, PA 15260, USA. [4] Key Lab of Mesoscopic Chemistry, School of Chemistry and Chemical Engineering, Nanjing University, 210093 Nanjing, China. Correspondence and requests for materials should be addressed to Y.Z. (email: zhuyan@nju.edu.cn) or to R.J. (email: rongchao@andrew.cmu.edu)

**N**ature has provided many elegant and precise orchestrations of hierarchical assembly with advanced and desired functionalities[1–4]. However, manipulating the hierarchical assembly of artificial nanoparticles[5–7] (NPs) remains one of the most challenging targets for nanoscientists, especially in terms of reaching a level of precision as high as that found in natural organisms[8–10]. Although impressive research progress has been achieved in the self-assembly of NPs in the past two decades[11–27], it is still difficult to access atomic-scale manipulation and characterization of the assemblies. Such insights are of critical importance for controlling and optimizing the functionality of NP assemblies[1,28]. Most conclusions from previous research are dependent on the analysis of the structure of assemblies by electron microscopy combined with theoretical calculations. This lacks direct evidence to illustrate some critical issues that strongly affect the self-assembly of NPs[28,29]. For example, the experimentally well-defined "picture", which can precisely depict how the surface ligands interact to drive the assembly, has only been rarely achieved[10], though the strategy of sticky ligands is widely applied to manipulate the self-assembly of NPs[30–32]. Meanwhile, it remains almost completely unknown how the associated counterions modulate the self-assembly of NPs[33,34], as it is extremely difficult to directly observe such small ions. Overall, these issues prevent researchers from directing the desired hierarchical assembly of NPs with tailored structure and functionality.

One feasible way to decipher these hidden codes is the X-ray diffraction study on the perfect NP-assembled single crystals[10]. The X-ray single-crystal study has the exclusive ability to directly reveal the atomic-level information of not only the total structure of NPs but also the interactions between interparticle ligands and the presence of associated counterions, which offers opportunities to pursue fundamental understanding of the properties of NP assemblies and establish definitive structure–property relationships. The prerequisite to obtain the perfect NP-assembled single crystal is to achieve the synthesis of atomically precise NPs, that is, every NP in the sample possesses absolutely the same formula and atomic structure, such that they can be treated as giant molecules[35–38].

Herein a hierarchical fibrous assembly of 21-gold-atom NPs ($Au_{21}$ for short hereafter) is realized and further modulated with atomic precision via tailoring the surface ligands and associated counterions. The atomic structure of this hierarchical assembly is fully revealed by single-crystal X-ray diffraction studies. It is found that the $Au_{21}$ NPs are first assembled into one-dimensional (1D) nanofibrils via π–π, anion–π, and aryl C-H⋯Cl interactions enabled by site-specific surface hooks, and in a subsequent process, the resultant 1D nanofibrils are further assembled into 3D crystals, giving rise to the hierarchical complexity (Fig. 1, an overview figure). Interestingly, through tailoring of the associated

counterions, this fibrous assembly of Au NPs can be exquisitely modulated, which significantly changes the electrical transport properties of the self-assembled solids by two orders of magnitude. We find that such a notable change of electric conductivity arises from the altered configurations of the interacting π–π pairs of the surface hooks, which are composed of four phenyl ligands and associated counterion. Such an atomic-level modulation of NP assembly via tailoring of ligands and counterions constitutes a major advance in pushing the NP assembly to ultimate precision.

## Results

**Tailoring the surface hooks on three atomically precise Au NPs.** The atomically precise $Au_{21}$ NPs with two different kinds of associated counterions are used as the building blocks in this work. The first building block is $[Au_{21}(SR)_{12}(PCP)_2]^+[AgCl_2]^{-39}$ (where $[AgCl_2]^-$ is the counteranion, R is cyclohexyl, and PCP is bis(diphenylphosphinomethane), IUPAC name: DPPM), which was obtained by site-specific tailoring of the surface of $[Au_{23}(SR)_{16}]^-[TOA]^{+40}$ (where TOA is tetraoctylammonium). The second building block is $[Au_{21}(SR)_{12}(PCP)_2]^+[Cl]^-$, which is newly synthesized in this work (details in the experimental, mass spectrum in Supplementary Fig 1). Figure 2a shows the atomic structures of $[Au_{23}(SR)_{16}]^-[TOA]^{+40}$, $[Au_{21}(SR)_{12}(PCP)_2]^+[AgCl_2]^{-39}$, and the new $[Au_{21}(SR)_{12}(PCP)_2]^+[Cl]^-$ determined by single-crystal X-ray analysis. All the three NPs possess the same 15-atom bipyramidal Au core with only a minor difference in the surface, i.e., in the two types of $[Au_{21}(SR)_{12}(PCP)_2]^+$ NPs, the two RS-Au-SR motifs (white circled in Fig. 2a, b) are replaced by two PCP motifs; Of note, there are two phenyl rings on each P atom (Fig. 2b). Interestingly, the different counteranions of the two $Au_{21}$ NPs, i.e., $[AgCl_2]^-$ and $[Cl]^-$, are found to reside near the phenyl ligands in the crystal structure (Fig. 2b). This is due to the anion–π and aryl C-H⋯Cl interactions (to be discussed in the following section). Based on the above results, the two $Au_{21}$ NPs can be treated as being connected by two more hooks, which are composed of four phenyl ligands and one associated counteranion (Fig. 2c), compared with the starting $Au_{23}$ NP, which has no such surface hooks.

**Hierarchical assembly of Au NPs directed by the tailored surface hooks.** Self-assembly of Au NPs into 3D single crystals was carried out by diffusing a non-solvent (pentane) into a $CH_2Cl_2$ (DCM) solution of the NPs (pentane:DCM = 10:1, volume). All the three types of NPs are readily assembled into large crystals with dimensions on the order of hundreds of μm (Fig. 3). The packing behaviors of the three types of Au NPs in their single crystals are fully revealed by X-ray analysis, which are found to be distinctly different from each other. For the

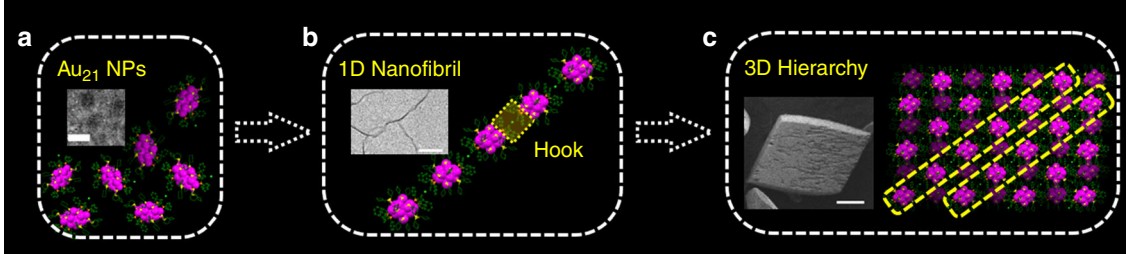

**Fig. 1** Schematic diagram of the hierarchical fibrous (1D to 3D) assembly of $Au_{21}$ NPs. Inserted graphs are the transmission electron microscopic (TEM) image of individual $Au_{21}$ NPs in **a** (Scale bar: 2 nm); TEM image of the 1D nanofibril assembled from $Au_{21}$ NPs in **b** (Scale bar: 50 nm); and the scanning electron microscopic (SEM) image of the 3D hierarchical crystals assembled from $Au_{21}$ NPs in **c** (Scale bar: 50 μm). Magenta = Au, light green = Cl, yellow = S, orange = P, green = C; H atoms are omitted for clarity. The yellow rectangle in **b** indicates the surface hook. The yellow rectangles in **c** are the 1D motifs (fibrils) in the 3D crystals

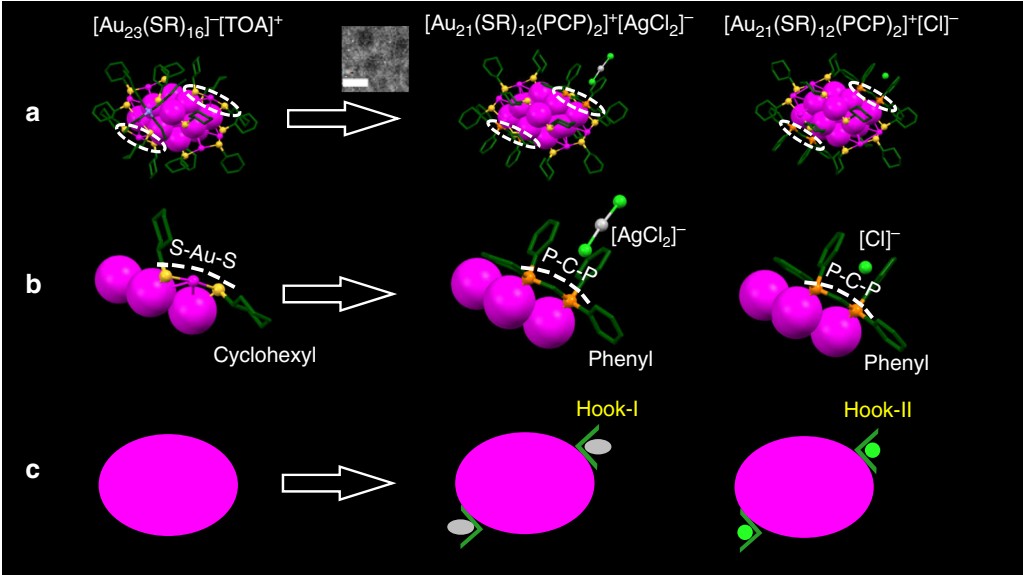

**Fig. 2** Tailoring the surface hooks on atomically precise Au NPs. **a** Atomic structures of the $[Au_{23}(SR)_{16}]^-[TOA]^+$, $[Au_{21}(SR)_{12}(PCP)_2]^+[AgCl_2]^-$, and $[Au_{21}(SR)_{12}(PCP)_2]^+[Cl]^-$. Inset is a TEM image of individual $Au_{21}$ NPs, scale bar: 2 nm. Circles are the different surface motifs. **b** Site-specific tailoring of the surface motifs and associated counterions of Au NPs: the two RS-Au-SR (R is cyclohexyl) surface motifs in $Au_{23}$ are replaced by two PCP motifs in $Au_{21}$; on a note, two phenyl rings are on each P atom. Dashed lines indicate the motifs. **c** Surface hooks on the Au NPs. Magenta = Au, gray = Ag, yellow = S, orange = P, green = C, light green = Cl, blue = N, all H atoms are omitted for clarity. The Au atoms in the bipyramidal core are displayed in space-filling mode. Surface Au, S, P, Cl, and N atoms are shown in ball-and-stick mode. The C atoms are shown in capped stick mode

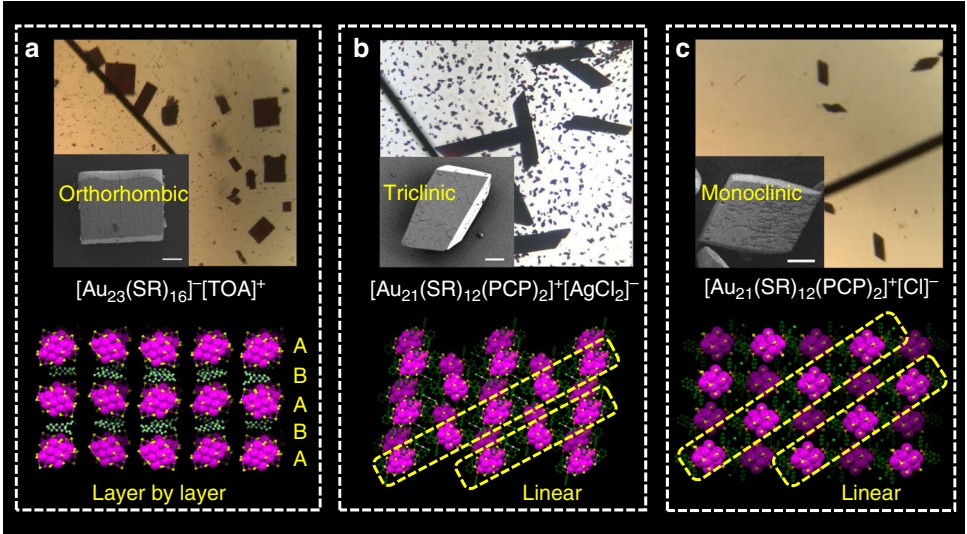

**Fig. 3** Hierarchical 3D crystals of the three types of Au NPs. Optical micrographs of the 3D single crystals of $[Au_{23}(SR)_{16}]^-[TOA]^+$ (**a**), $[Au_{21}(SR)_{12}(PCP)_2]^+[AgCl_2]^-$ (**b**), and $[Au_{21}(SR)_{12}(PCP)_2]^+[Cl]^-$ (**c**) are displayed. Insets are the SEM micrographs of the 3D crystals, scale bar: 50 μm. Packing of the three types of Au NPs in their single crystals revealed by X-ray diffraction analysis. The $[Au_{23}(SR)_{16}]^-[TOA]^+$ crystal grows in a layer-by-layer manner (ABAB…) with $[Au_{23}(SR)_{16}]^-$ in layer A and the counterion $[TOA]^+$ in layer B (**a**). In the crystals of the two $Au_{21}$ NPs (**b**, **c**), the NPs are linearly assembled (yellow circles)

$[Au_{23}(SR)_{16}]^-[TOA]^+$, the shape of the single crystal is rectangular, and the NPs are packed into a base-centered orthorhombic lattice (Fig. 3a). On the other hand, in the single crystals of $[Au_{21}(SR)_{12}(PCP)_2]^+[AgCl_2]^-$, the NPs are packed into a triclinic lattice and the shape of the single crystal is a parallelogram (Fig. 3b). The packing mode of $[Au_{21}(SR)_{12}(PCP)_2]^+[Cl]^-$ in the single crystal is also different from the previous two NPs, which shows a monoclinic lattice (Fig. 3c). Interestingly, taking a closer look at the arrangement of NPs, one can find that the $[Au_{23}(SR)_{16}]^-[TOA]^+$ crystal grows in a layer-by-layer manner (ABAB…) with $[Au_{23}(SR)_{16}]^-$ in layer A and the counterion

$[TOA]^+$ in layer B. However, the $Au_{21}$ NPs show a totally different packing mode in which each type of $Au_{21}$ (associated with either $[AgCl_2]^-$ or $Cl^-$) is linearly assembled along the diagonal of the {100} plane in the single crystal of $[Au_{21}(SR)_{12}(PCP)_2]^+[AgCl_2]^-$ and along the diagonal of the {010} plane in the $[Au_{21}(SR)_{12}(PCP)_2]^+[Cl]^-$ case.

Figure 4 shows the details of the 1D self-assembly of each type of $Au_{21}$ NPs in their single crystals, which shows the distinct effects of the counterions. The linear arrangement in the $Au_{21}$ NP crystals is driven by specific interparticle interactions induced by the phenyl ligands of the surface hooks. Indeed, when zooming in

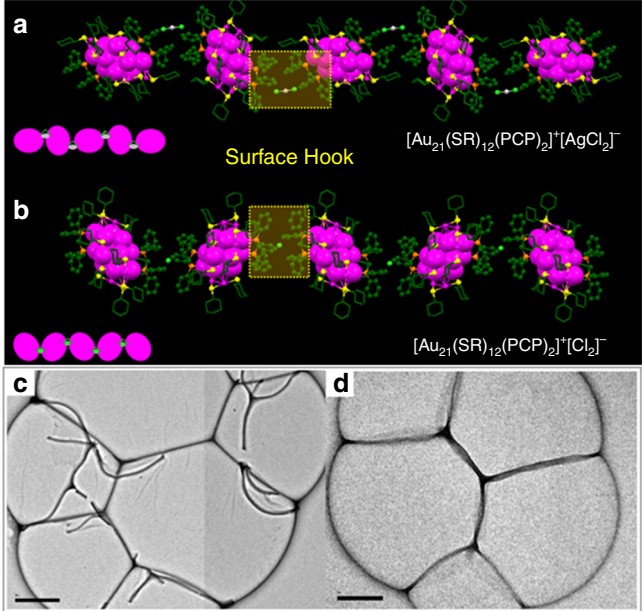

**Fig. 4** 1D Nanofibrils assembled from the two types of $Au_{21}$ NPs. **a**, **b** are the packing of $[Au_{21}(SR)_{12}(PCP)_2]^+[AgCl_2]^-$ and $[Au_{21}(SR)_{12}(PCP)_2]^+[Cl]^-$ in their 1D assemblies. The orientation of Au NPs is modulated by the counterion. Magenta = Au, gray = Ag, light green = Cl, yellow = S, orange = P, green = C. All H atoms are omitted for clarity. Yellow areas are the surface hooks connecting neighboring NPs. **c**, **d** display the TEM images of the 1D self-assembly of $[Au_{21}(SR)_{12}(PCP)_2]^+[AgCl_2]^-$ and $[Au_{21}(SR)_{12}(PCP)_2]^+[Cl]^-$, respectively, scale bar: 200 nm

the surfaces of the two neighboring $Au_{21}$ NPs, the different packing can be correlated to the alignment of surface phenyl ligands from neighboring NPs. As shown in Fig. 4a, b, the two groups of phenyl ligands from two neighboring NPs are approaching each other, acting as hooks which link the two neighboring NPs and direct their self-assembly. Additionally, the counter anions ($[AgCl_2]^-$ and $[Cl]^-$) are found to reside between the two batches of phenyl ligands from the two neighboring NPs, forming an "anion in the phenyl cage" superstructure. It can be observed that different counteranions influence the arrangement of phenyl ligands, hence changing the orientations of NPs in the assembly. Interestingly, this 1D assembly can be experimentally realized under the same experimental conditions but with much less non-solvent (pentane) diffused into the DCM solution of the NPs (pentane:DCM = 1:1), as shown in Fig. 4c, d by transmission electron microscopic (TEM) imaging.

We further zoom in the crystal structure to investigate the interaction details between the hooks of the neighboring $Au_{21}$ NPs (Fig. 5). It has previously been demonstrated that the π–π interactions can strongly influence the higher-order structures of biomacromolecules and the crystal packing of organic molecules[41–43]. Generally, the interacting π–π pairs are found in a wide range of sandwich, T-shape, and parallel-displaced arrangements with distances ranging from 4.5 to 7 Å to be most common for the two phenyl rings[41–43]. In the $[Au_{21}(SR)_{12}$ $(PCP)_2]^+[AgCl_2]^-$ crystal, the pair of phenyl rings from neighboring NPs are found to be arranged in optimal or distorted T-shape configurations, with distances ranging from ~5.0 to ~5.8 Å (Fig. 5a). In the $[Au_{21}(SR)_{12}(PCP)_2]^+[Cl]^-$ crystal, parallel-displaced configuration is observed, with distance of ~5.7 Å between the phenyl pairs (Fig. 5b). In both $Au_{21}$ crystals, the sandwich configuration of phenyl pairs can also be clearly identified with a distance of ~3.9 Å in each PCP motif from the same NP. It should be noted that the relatively long distance of

the π–π pairs in this work is similar to the distances found in proteins with aromatic side chains, though not to the distances in small molecule crystal structures. On the other hand, the two small counteranions $[Cl]^-$ and $[AgCl_2]^-$ are found to reside only in the phenyl cage, suggesting interactions between the counteranion and phenyl rings. The average distance between the centroid of the aromatic ring and the anion is determined to be ~4.9 Å with angles ranging from 0° to 45°. Similar geometric parameters have been observed in proteins and nucleic acids[44]. Futhermore, in both $Au_{21}$ NP crystal structures, the $Cl^-$ is surrounded by several C-H···Cl interactions ranging from 2.5 to 3.3 Å. Although weak in nature, these interactions may certainly add up to the above-mentioned π-type interactions[45].

Overall, our results directly reveal a two-step hierarchical fibrous assembly of $Au_{21}$ NPs and identify the directing effect of the surface hooks. It can be concluded that the $Au_{21}$ NPs (each type) are first assembled into 1D nanofibrils. This anisotropic process is driven by the site-specific surface hooks that provide directional π–π, anion–π, and aryl C-H···Cl interactions and the atomic configurations of the surface hooks are experimentally defined. In a subsequent process, the 1D nanofibrils are further assembled into hierarchical 3D crystals. A quasi close-packing manner is identified in the 1D to 3D process (Supplementary Fig 2 and 3), which indicates the entropy-driven mechanism for this process. In contrast, for $[Au_{23}(SR)_{16}]^-$ NPs that has no such surface hooks, they show a totally different 2D to 3D assembly behavior. It can be observed that 2D nanosheets were first assembled under less amount of the nonsolvent: pentane/DCM = 1:1 (TEM images in Supplementary Fig 4), and in a subsequent process, the 2D nanosheets further stacked into hierarchical 3D crystals (Supplementary Fig 5).

**Modulating the electron transport properties in $Au_{21}$ NP assembled solids by the surface hooks.** The differences in the crystal packing of the two types of $Au_{21}$ NPs and their associated different counterions ($[AgCl_2]^-$ and $Cl^-$) result in distinctive electrical transport properties. To measure the charge transport properties, we fabricated two-terminal devices by painting silver contacts on individual single crystals. The room temperature conductivity is extracted from the slope of the linear I–V curve (Fig. 6a) averaged over 6–7 single crystal devices (Supplementary Fig 6 and 7). The average electrical conductivity ($\sigma$) of $[Au_{21}(SR)_{12}(PCP)_2]^+[AgCl_2]^-$ ($\sigma$~$1.44\times10^{-8}$ S/m) is approximately two orders of magnitude smaller than that of $[Au_{21}(SR)_{12}(PCP)_2]^+[Cl]^-$ ($\sigma \sim 2.38\times10^{-6}$ S/m). Higher conductivity in a NP assembly is often attributed to a smaller interparticle spacing. The average center-to-center distances of neighboring $Au_{21}$ NPs in the $[AgCl_2]^-$ and $[Cl]^-$ structures are 16.80 and 16.39 Å, respectively. The significantly different conductivity (by two orders of magnitude) with such a small change of interparticle distance (0.41 Å) contrasts with previous experimental results of the conductivity measured from Au NP films[46–49] (~1 order of magnitude for every 1 Å increase of interparticle distance). This suggests that, besides the well-known interparticle distance factor, other effects could also be responsible for the large difference in electron transport in these NP crystals. In previous work, this was unable to be further analyzed owing to the lack of atomic-level characterization. Based on the typical hopping transport mechanism for weakly coupled NPs illustrated in Fig. 6b[46], conductivity $\sigma$ can be expressed as $\sigma = \exp(-\beta d)$, where $d$ is the interparticle distance and $\beta$ is the tunneling decay constant. Note that the tunneling decay constant is dependent on the barrier height, which is strongly affected by the nature and interactions of ligands (Fig. 6b). Previous studies of organic materials[50,51] show that the electron transport is sensitive to the configurations of the π–π pairs and it

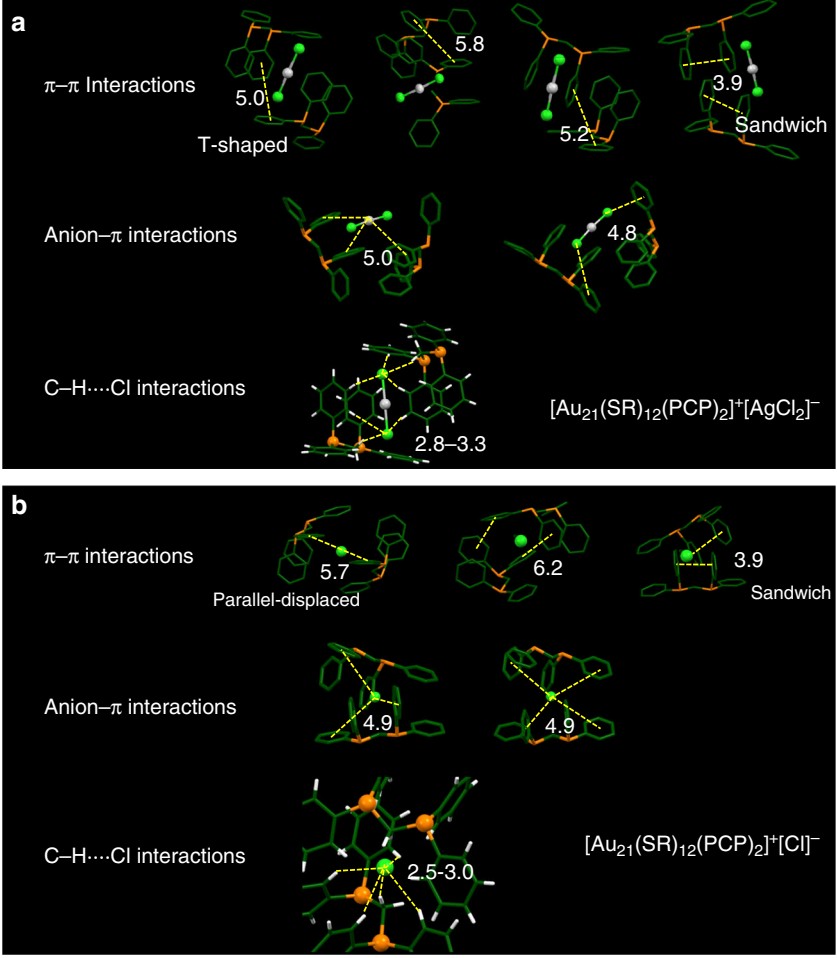

**Fig. 5** Anatomy of the interactions of surface hooks. Illustration of the interparticle π–π, anion–π, and C-H··Cl interactions for **a** $[Au_{21}(SR)_{12}(PCP)_2]^+$ $[AgCl_2]^-$ and **b** $[Au_{21}(SR)_{12}(PCP)_2]^+[Cl]^-$. Gray = Ag, yellow = S, orange = P, green = C, light green = Cl, white = H. The yellow dashed lines correspond to the measured distances

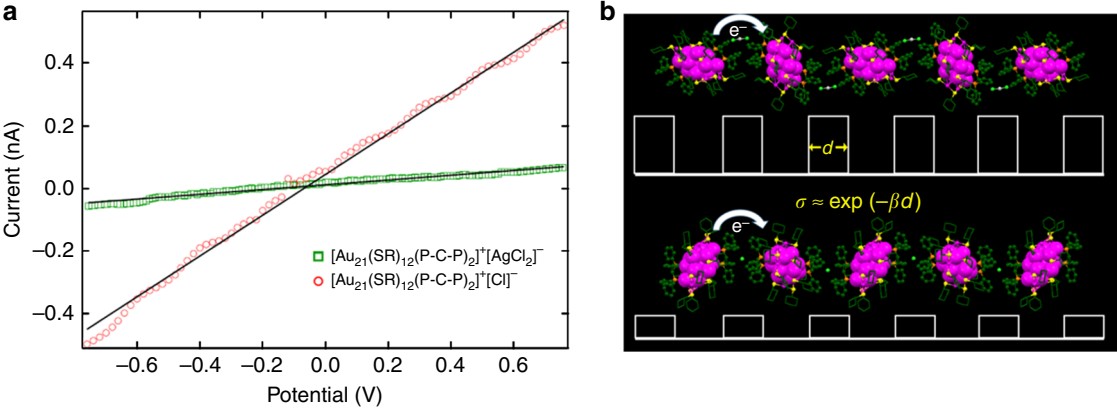

**Fig. 6** Electron transport properties of the Au$_{21}$ NP assemblies. **a** Room temperature conductivity of the single crystals of $[Au_{21}(SR)_{12}(PCP)_2]^+[AgCl_2]^-$ (green) and $[Au_{21}(SR)_{12}(PCP)_2]^+[Cl]^-$ (red), respectively. **b** Schematic diagram of the electron hopping in Au$_{21}$ NP assemblies. Different configurations of the interacting π–π pairs result in different height of tunneling barriers (white solid squares), thus changing the electron conductivity (e$^-$ represents the electron; σ is the conductivity; d is the interparticle distance, and β is the tunneling decay constant)

was found that the parallel-displaced π-stacking arrangement of the molecules is the best for electron conductance. This is consistent with our results that, in the [Cl]$^-$ crystal, such a parallel-displaced π-stacking is observed (Fig. 5b), which leads to a lower barrier and

hence higher conductivity. To our knowledge, our work is the first to provide an atomic-level experimental demonstration of the major role of counterion in modulating the interactions of phenyl ligands

and tailoring the height of the tunneling barrier, which in turn modulates the electron transport in NP assemblies.

## Discussion

In conclusion, a hierarchical fibrous assembly of Au NPs is realized and further modulated with atomic precision via tailoring the surface "hook," which is composed of four phenyl rings associated with a specific counterion. This work presents an experimentally well-defined, atomic-level answer to the long-time discussed question, that is, how "sticky" ligands and associated counterions could guide the self-assembly of NPs. Especially, we have demonstrated the power of the counterion in exquisitely modulating the structure and property of hierarchical assemblies. This work offers a new insight into the structural factors for controlling the electron transport in NP assemblies and enhances our abilities to create new hierarchical NP assemblies with desired structures and properties.

## Methods

**Experimental**. $[Au_{23}(SR)_{16}]^{-}[TOA]^{+}$ and $[Au_{21}(SR)_{12}(PCP)_2]^{+}[AgCl_2]^{-}$ were synthesized and crystallized by previously reported methods[39,40]. To synthesize the new $[Au_{21}(SR)_{12}(PCP)_2]^{+}[Cl]^{-}$, 20 mg $[Au_{23}(SR)_{16}]^{-}[TOA]^{+}$ and 20 mg $[Au_{21}(SR)_{12}(PCP)_2]^{+}[AgCl_2]^{-}$ were first dissolved in 3 ml DCM and then allowed to react for ~3 h at room temperature, which led to the $[Au_{21}(SR)_{12}(PCP)_2]^{+}[Cl]^{-}$ product. To grow 3D single crystals, DCM solution containing the Au NPs was transferred to a glass tube and pentane (~10 times DCM) was diffused into the solution at room temperature for ~2 days. To achieve the 1D self-assembly of each of the two types of $Au_{21}$ NPs, the amount of pentane was reduced to ~1:1 (pentane/DCM). TEM measurements were performed on a JEOL-2000EX microscope operating at 200 kV. Scanning electron microscopy (SEM) was conducted on a ZEISS Sigma 500 VP SEM microscope. Details of the X-ray crystallographic analysis are provided in Supplementary Methods.

**Electrical transport measurements**. Single crystals of the $[Au_{21}(SR)_{12}(PCP)_2]^{+}[AgCl_2]^{-}$ and $[Au_{21}(SR)_{12}(PCP)_2]^{+}[Cl]^{-}$ NPs were selected and adhered to a glass slide. Contacts were made by painting to opposite sides of the crystal with conductive silver paint (Ted Pella 16032). $I$–$V$ curves were collected on a probe station (Agilent Semiconductor Parameter Analyzer 4155C). A linear fit was applied to the $I$–$V$ curves and resistance was extracted. Resistivity was calculated by measuring the dimensions of each sample with an optical microscope.

## Data availability

The X-ray crystallographic coordinates for structure reported in this work (Supplementary Table 1) have been deposited at the Cambridge Crystallographic Data Centre (CCDC) under deposition number CCDC-1861291. These data can be obtained free of charge from The Cambridge Crystallographic Data Centre via www.ccdc.cam.ac.uk/data_request/cif.

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

## Acknowledgements

R.J. acknowledges the financial support from the AFOSR under AFOSR Award No. FA9550-15-1-9999 (FA9550-15-1-0154). The electrical transport measurements were supported by the AFOSR under AFOSR Award No. FA9550-18-1-0020. J.C.R is supported by the Department of Defense NDSEG Fellowship. Y.Z. acknowledges financial support from the National Natural Science Foundation of China (21773109).

## Author contributions

Q.L., R.J. and Y.Z. designed the research; Q.L. performed the synthesis and self-assembly experiments; J.C.R. and X.R. conducted the electrical transport measurements. T.-Y.L. and N.L.R conducted X-ray crystallographic analysis. All authors analyzed data and contributed to the writing of the manuscript.

## Additional information

**Competing interests:** The authors declare no competing interests.

