## [Peer Review File · Nature Communications]

Reviewers' comments:

Reviewer #1 (Remarks to the Author):

I do not recommend this paper for publication. The claim that the crystallographic results "directly reveal an elegant hierarchical fibrous assembly of Au₂₁ NPs in the crystals and unambiguously identify the corresponding driving forces" is overstated. First, the use of the "hierarchical fibrous assembly" is a fancy way of saying that the crystal (which is always an assembly) exhibits a one-dimensional motif (which often occurs). Secondly, the driving force is attributed to long-range interactions involving $\pi\cdots\pi$ interactions and either [Cl-Au-Cl]⁻ or Cl⁻ ions. The fact that they quote reference 39 in stating that these $\pi\cdots\pi$ interactions occur with "distances ranging from 4.5 to 7 Å" shows that they didn't read the rest of the paper. These distances are found in proteins with aromatic side chains. But in normal crystal structures, the $\pi\cdots\pi$ stacking is generally in the range of 3.4 -3.8 Å, and edge-to-face interactions around 5 Å. Parallel-displaced interactions are ca. 3.2 – 3.8 Å. Even at these distances, the energy involved is "structure directing" but not anywhere near the strength of a normal bond. Few of the "hooks" shown in Figure 5 look to be very significant.

The modification of the surface of the NP with "PCP" to yield the monoclinic form is a nice piece of work. However, as shown in Figure 3, a linear arrangement can be identified in the orthorhombic and triclinic Au NP's as well.

If there is something special about the monoclinic form, then the electrical transport should be tested along specific, indexed, crystal directions, especially those that parallel the onedimensional stacks. I didn't see any indication that more than one direction was tested.

My biggest criticism is that the crystal structure has omitted the two hydrogen atoms that would be bonded to the atom C1 (middle of the "PCP"). When those two hydrogens are included, the Cl⁻ is surrounded by 10 C-H...Cl interactions ranging from 2.67 to 3.03 Å. Although weak in nature, these interactions may certainly add up to the above mentioned π -type interactions. These interactions are well-accepted by crystallographers in their importance. Just one example is the recent work by Diop, et al. in Acta Cryst. Section E, **2018**, 74, 69. Therefore, the driving force, what is called the "hooks" is not at all unambiguous.

A few other issues:

1. Figure 1 caption: the C atoms are not omitted—they are shown in green.
2. "PCP" is named "dppm" by the IUPAC.
3. on p. 3 it is stated that "PCP is bis(diphosphinomethane)" but it is actually bis(diphenylphosphinomethane).
4. Figure 2, the caption should explain the significance of the yellow dotted regions.
5. The abstract should at least state that the paper is about Au₂₁ NP's.
6. Why the Au₂₁ is called a bipyramid is mysterious!

Reviewer #2 (Remarks to the Author):

In this manuscript, the authors exquisitely designed three Au nanoclusters that possessed the same structure but minor difference in the surface. Interestingly, these three clusters displayed quite different packing behaviors both in 3-D and 1-D self-assemblies. Moreover, the different crystal packing demonstrates the distinctive electrical transport properties. The mechanism about the surface and counter anion dependent self-assembly behaviors were elucidated clearly by X-ray diffraction study. Thus, based on its novelty and importance, the manuscript is strongly recommended for publication in Nature communication after minor revisions.

1. The language should be polished throughout the context.
2. The TEM images of 1D self-assembly of $[\text{Au}_{21}(\text{SR})_{12}(\text{PCP})_2] + [\text{Cl}]^-$ were not shown in text. The authors needs to add some pictures to enrich this work.
3. There is an error in the method part for synthesis of $[\text{Au}_{23}(\text{SR})_{16}] - [\text{TOA}]^+$, $[\text{Au}_{21}(\text{SR})_{12}(\text{PCP})_2] + [\text{AgCl}_2]^-$ and $[\text{Au}_{21}(\text{SR})_{12}(\text{PCP})_2] + [\text{Cl}]^-$. Please check carefully and add more detailed information.
4. To enhance the comprehensiveness, some important and closely related literatures need to be cited, e.g. Nature nanotechnology, 2011, 6, 580-587; Nature Communications 2017, 8, 14038.

Reviewer #3 (Remarks to the Author):

This manuscript authored by Jin et. al. reports a new strategy to assemble Au nanoparticles (21-gold-atom clusters) into a unique hierarchical fibrous structure via tailoring the interaction between surface ligand and counterions. Constructing nanoparticles assembly with predictable and controllable structures and atom-accuracy ordering stands as one of the greatest challenge in materials science. Two types of Au nanoparticles: $[\text{Au}_{21}(\text{SR})_{12}(\text{PCP})_2] + [\text{AgCl}_2]^-$ and $[\text{Au}_{21}(\text{SR})_{12}(\text{PCP})_2] + [\text{Cl}]^-$ are studied in the present work. The PCP ligand and anions (AgCl_2^- or Cl^-) on nanoparticle surface exhibit strong π - π and anion- π interactions which can be used as surface hooks to direct the formation of 1-D assembly. The assembled structure is carefully studied by single crystal XRD. The authors also studied the electrical transport properties of the two types of nanoparticles assemblies and found that the one with Cl^- anion shows higher conductivity. It is a very well-written manuscript and a good addition to the materials design methods with complex structures, which should be disseminated to the broad audience. I have a few of minor comments:

[1] The authors should provide the monodispersity information of $[\text{Au}_{21}(\text{SR})_{12}(\text{PCP})_2] + [\text{AgCl}_2]^-$ and $[\text{Au}_{21}(\text{SR})_{12}(\text{PCP})_2] + [\text{Cl}]^-$ they synthesized. MALDI-MS data should be added. More details should be included, for example, any size selection process is used?

[2] For electrical transport measurement, conductivity deviation and error bar for each assembly in different testing should be provided and commented.

[3] It is interesting to see the two orders of magnitude conductivity difference between two types of assemblies with different anions. The authors claim that this conductivity difference is not related to a small change of interparticle distance between two assemblies (0.41 Å), as the previous results of the conductivity measured from Au NP films in ref 44-46 only exhibit 1 order of magnitude change for every 1 Å increase of interparticle distance. But considering both conductivity values in the present

work are very low (10^{-6} to 10^{-8} S/m, can be treated as insulator), this conclusion may be questionable. The 1 order of magnitude conductivity change with 1 Å interparticle distance increase in ref 44-46 is observed in the range of 10^{-1} to 10^{-3} S/m where 1 order of value difference is very dramatic. Comparing to this, the conductivity change in the present work is not as significant as claimed, and it is not convincing to rule out the interparticle distance effect. The authors should comment more on this point.

We thank all the reviewers for very careful reading of our manuscript and providing insightful comments for us to improve our manuscript. Below is our point-by-point response to reviewers' comments (response in blue, revision in red).

Reviewer #1, pages 1-5

Reviewer #2, pages 6-7

Reviewer #3, pages 8-9

Reviewer #1 (Remarks to the Author):

I do not recommend this paper for publication. The claim that the crystallographic results "directly reveal an elegant hierarchical assembly of Au₂₁ NPs in the crystals and unambiguously identify the corresponding driving forces" is overstated.

First, the use of the "hierarchical fibrous assembly" is a fancy way of saying that the crystal (which is always an assembly) exhibits a one-dimensional motif (which often occurs).

Response: Thanks for the comments. To clarify, we used the phrase "hierarchical fibrous assembly" not because of the one-dimensional motif that can often be found in the crystal; rather, because the final 3D crystal is actually assembled by the 1D NP-assembled fibers. As we experimentally demonstrated, for the $[\text{Au}_{21}(\text{SR})_{12}(\text{PCP})_2]^+[\text{AgCl}_2]^-$, the NPs first assemble into one-dimensional (1D) nanofibrils under small amounts of nonsolvent: pentane (1:1), then these 1D nanofibrils further assemble into the 3D crystals under excess pentane. This kind of "isolated NP – 1D nanofibril – 3D crystal" assembly mode was also observed in the other system, $[\text{Au}_{21}(\text{SR})_{12}(\text{PCP})_2]^+[\text{Cl}]^-$ (see revised Figure 1). In contrast, for the Au₂₃ which has no such two hooks, it shows a mode of "isolated NP – 2D nanosheet – 3D crystal" (newly added in Figure S4), that is, the Au₂₃ NP first assemble into 2D nanosheets under small amounts of nonsolvent – pentane (1:1). Therefore, our intention with the use of "hierarchical fibrous assembly" is to describe the 1D to 3D directed hierarchical interactions. To avoid confusion, we have explained the "hierarchical fibrous assembly" as "hierarchical "1D to 3D" assembly" for the two Au₂₁ in the abstract and use "hierarchical "2D to 3D" assembly" to describe the assembly behavior of Au₂₃ in the revised manuscript. We hope this is now acceptable.

Revision: revised Figure 1, newly added Figures S4 and S5.

Figure 1. Hierarchical fibrous assembly of Au NPs. Magenta: Au, light green: Cl, yellow: S, orange: P, green: C; H atoms are omitted for clarity.

Figure S4. Two-dimensional nanosheets assembled from $[\text{Au}_{23}(\text{SR})_{16}]^{-}[\text{TOA}]^{+}$ under DCM/Pentane = 1:1.

Figure S5. The “2D to 3D” hierarchical assembly of $[\text{Au}_{23}(\text{SR})_{16}]^{-}[\text{TOA}]^{+}$.

Revision (in the Abstract):

“..... In this work, a hierarchical fibrous (two-step 1D to 3D) assembly of Au NPs (21-gold atom, Au_{21}) is realized....”

Secondly, the driving force is attributed to long-range interactions involving $\pi \dots \pi$ interactions and either $[\text{Cl-Au-Cl}]^{-}$ or Cl^{-} ions. The fact that they quote reference 39 in stating that these $\pi \dots \pi$ interactions occur with "distances ranging from 4.5 to 7 Å" shows that they didn't read the rest of the paper. These distances are found in proteins with aromatic side chains. But in normal crystal structures, the $\pi \dots \pi$ stacking is generally in the range of 3.4 -3.8 Å, and edge-to-face interactions around 5 Å. Parallel-displaced interactions are ca. 3.2 – 3.8 Å. Even at these distances, the energy involved is "structure directing" but not anywhere near the strength of a normal bond. Few of the "hooks" shown in Figure 5 look to be very significant.

Response: We thank the reviewer for the insightful comments. This long range of distances of π pairs can also be identified in the other atomically-precise Au NP crystals such as the single crystal of Au_{25} (see the following figure, ref. *J.Am.Chem. Soc.*, **2008**, *130*, 5883–5885). This is a very interesting issue;

however, it didn't attract attention in our research community and wasn't discussed in previous papers. Actually, the molecular weight of the Au₂₁ NP is large (~6288 Da), thus it is a large molecule similar to some proteins. Besides, the ligands on the surface of Au₂₁ NP are also comparable to the side chains of protein as their motion is also restricted. Thus, it is reasonable that the observed distances of our π pairs are more like those found in proteins rather than small molecule engineering.

Figure (for Response). The π - π interactions in the Au₂₅ single crystal.

We agree that the "structure directing" is more accurate and a revision has been made. The convincing evidence of the "structure directing" effect of the π pair hooks is that for the Au₂₃ which has no such hooks, its assembly behavior is totally different from the two Au₂₁ systems (change from "1D to 3D" to "2D to 3D"), as shown in the newly added Figures S4 and S5.

Revision: new comments added in pages 7 and 8:

..... It should be noted that the relatively long distance of the π - π pairs in this work is similar to the distances found in proteins with aromatic side chains, though not to the distances in small molecule crystal structures.

..... Overall, our results directly reveal a two-step hierarchical fibrous assembly of Au₂₁ NPs and identify the "directing effect" of the "surface hooks".

..... In contrast, for [Au₂₃(SR)₁₆]⁻ NPs which has no such surface hooks, they show a totally different 2D to 3D assembly behavior. It can be observed that 2D nanosheets were first assembled under less amounts of nonsolvent: pentane/DCM = 1:1 (TEM images in Figure S4) and in a subsequent process, the 2D nanosheets further stacked into hierarchical 3D crystals (Figure S5).

The modification of the surface of the NP with "PCP" to yield the monoclinic form is a nice piece of work. However, as shown in Figure 3, a linear arrangement can be identified in the orthorhombic and triclinic Au NP's as well. If there is something special about the monoclinic form, then the electrical transport should be tested along specific, indexed, crystal directions, especially those that parallel the onedimensional stacks. I didn't see any indication that more than one direction was tested.

Response: Thanks for the excellent comments. Unfortunately, the small size and extreme fragility of our crystals made the control of crystallographic direction very challenging. We had no success in performing the suggested measurement. If larger crystals can be grown, mapping the behavior along specific directions would be a compelling future experiment.

My biggest criticism is that the crystal structure has omitted the two hydrogen atoms that would be bonded to the atom C1 (middle of the "PCP"). When those two hydrogens are included, the Cl⁻ is surrounded by 10 C-H...Cl interactions ranging from 2.67 to 3.03 Å. Although weak in nature, these interactions may certainly add up to the above mentioned π -type interactions. These interactions are well-accepted by crystallographers in their importance. Just one example is the recent work by Diop, et al. in

Acta Cryst. Section E, **2018**, 74, 69. Therefore, the driving force, what is called the "hooks" is not at all unambiguous.

Response: Thanks for the reviewer's careful reviewing and excellent suggestion. We have followed the suggestion and revised our manuscript accordingly. We agree that C-H...Cl interactions exist, and the corresponding C-H...Cl interactions are also part of the "hook" effect. But it should be noted that there are other 12 cyclohexanethiol ligands on the surface of Au₂₁, each has 6 C-H bond and thus theoretically, the C-H...Cl can also form between these H atoms and Cl. The X-ray experimental results show that Cl only reside in the "phenyl cages"; this suggests that both the anion-π interactions (which we have already mentioned) and the aryl C-H...Cl interactions (newly added in the revised manuscript) determine the position of the anion Cl⁻ in the crystals.

Revision: (newly added comments on page 7):

"... Furthermore, in both two Au₂₁ NP crystal structures, the Cl⁻ is surrounded by C-H...Cl interactions ranging from 2.5 to 3.3 Å. Although weak in nature, these interactions may certainly add up to the above mentioned π-type interactions.^[45]"

[45] M. B. Diop, L. Diop, A. G. Oliver, Acetyltri-phenyl-phospho-nium 2,3,5-tri-phenyl-tetra-zolium tetra-chlorido-cuprate(II) *Acta Crystallographica Section E* **74**, 69-71 (2018).

Also, in the Abstract, we have revised as follows:

"The key instruction for this hierarchical fibrous assembly is encoded in a site-specifically tailored "hook" on the NP surface (composed of four phenyl ligands associated with a counteranion), which provides anisotropic and directional π-π, anion-π and aryl C-H...Cl interactions for assembly of NPs."

The mentioned ref has also been added:

Revised Figure 5:

Figure 5. Anatomy of the interactions of surface "hooks". Illustration of the interparticle π-π, anion-π and C-H...Cl interactions for (A) Au₂₁(SR)₁₂(PCP)₂⁺[AgCl₂]⁻ and (B) [Au₂₁(SR)₁₂(PCP)₂]⁺Cl⁻. Gray = Ag, yellow = S, orange = P, green = C, light green = Cl, white = H.

A few other issues:

1. Figure 1 caption: the C atoms are not omitted—they are shown in green.

Response: It has been corrected.

2. "PCP" is named "dppm" by the IUPAC.

Response: The IUPAC name "dppm" is noted in revised manuscript. We use PCP to help readers more easily understand its structure since P-C-P is more intuitive than dppm.

3. on p. 3 it is stated that "PCP is bis(diphosphinomethane)" but it is actually bis(diphenylphosphinomethane).

Response: It has been corrected. Thank you for catching the error!

4. Figure 2, the caption should explain the significance of the yellow dotted regions.

Response: More explanations have been added as suggested.

Revision (in the caption of Figure 2):

“..... Circled areas are the different surface motifs.”

5. The abstract should at least state that the paper is about Au₂₁ NP's.

Response: It has been revised as suggested.

Revision (in the Abstract):

“...In this work, a hierarchical fibrous assembly of Au NPs (21-gold atom, Au₂₁) is realized...”

6. Why the Au₂₁ is called a bipyramid is mysterious!

Response: The Au₂₁ has a 15-gold-atom bipyramid core, see the following figure:

Reviewer #2 (Remarks to the Author):

In this manuscript, the authors exquisitely designed three Au nanoclusters that possessed the same structure but minor difference in the surface. Interestingly, these three clusters displayed quite different packing behaviors both in 3-D and 1-D self-assemblies. Moreover, the different crystal packing demonstrates the distinctive electrical transport properties. The mechanism about the surface and counter anion dependent self-assembly behaviors were elucidated clearly by X-ray diffraction study. Thus, based on its novelty and importance, the manuscript is strongly recommended for publication in Nature communication after minor revisions.

1. The language should be polished throughout the context.

Response: We have corrected the language in the revised manuscript.

2. The TEM images of 1D self-assembly of $[\text{Au}_{21}(\text{SR})_{12}(\text{PCP})_2]^+[\text{Cl}]^-$ were not shown in text. The authors need to add some pictures to enrich this work.

Response: The TEM images of 1D self-assembly of $[\text{Au}_{21}(\text{SR})_{12}(\text{PCP})_2]^+[\text{Cl}]^-$ are added as suggested.

Revision: Revised Figures 1 and 4 in the revised manuscript.

Figure 1. Hierarchical fibrous assembly of Au NPs. Magenta: Au, light green: Cl, yellow: S, orange: P, green: C; H atoms are omitted for clarity.

(panels C and D: next page)

Figure 4. 1D Nanofibrils assembled from the two Au₂₁ NPs. (A) and (B) are the packing of [Au₂₁(SR)₁₂(PCP)₂]⁺[AgCl₂]⁻ and [Au₂₁(SR)₁₂(PCP)₂]⁺[Cl]⁻ in their 1D assemblies. The orientation of Au NPs is modulated by the counterion. Magenta: Au, gray: Ag, light green: Cl, yellow: S, orange: P, green: C. H atoms are omitted for clarity. Figure (C) and (d) display the TEM images of the 1D self-assembly of [Au₂₁(SR)₁₂(PCP)₂]⁺[AgCl₂]⁻ and [Au₂₁(SR)₁₂(PCP)₂]⁺[Cl]⁻, respectively.

3. There is an error in the method part for synthesis of [Au₂₃(SR)₁₆]⁻[TOA]⁺, [Au₂₁(SR)₁₂(PCP)₂]⁺[AgCl₂]⁻ and [Au₂₁(SR)₁₂(PCP)₂]⁺[Cl]⁻. Please check carefully and add more detailed information.

Response: The error has been corrected and more experimental details are added.

Revision (page 10):

“*Experimental* [Au₂₃(SR)₁₆]⁻[TOA]⁺ and [Au₂₁(SR)₁₂(PCP)₂]⁺[AgCl₂]⁻ were synthesized and crystallized by previously reported methods.^[11] To synthesize the new [Au₂₁(SR)₁₂(PCP)₂]⁺[Cl]⁻, 20 mg [Au₂₃(SR)₁₆]⁻[TOA]⁺ and 20 mg [Au₂₁(SR)₁₂(PCP)₂]⁺[AgCl₂]⁻ were first dissolved in 3 ml DCM and then allowed to react for ~3 hours at room temperature, which led to the [Au₂₁(SR)₁₂(PCP)₂]⁺[Cl]⁻ product. To grow 3D single crystals, DCM solution containing the Au NPs was transferred to a glass tube and pentane (~10 times DCM) was diffused into the solution at room temperature for ~2 days. To achieve the 1D self-assembly of each of the two types of Au₂₁ NPs, the amount of pentane was reduced to ~ 1 : 1 (pentane/DCM). TEM measurements were performed on a JEOL-2000EX microscope operating at 200 kV. SEM was conducted on a ZEISS Sigma 500 VP SEM microscope. Details of the x-ray crystallographic analysis are provided in the supporting information.”

4. To enhance the comprehensiveness, some important and closely related literatures need to be cited, e.g. Nature nanotechnology, 2011, 6, 580-587; Nature Communications 2017, 8, 14038.

Response: The suggested references are added.

Revision: New refs 26 and 27.

Reviewer #3 (Remarks to the Author):

This manuscript authored by Jin et. al. reports a new strategy to assemble Au nanoparticles (21-gold-atom clusters) into a unique hierarchical fibrous structure via tailoring the interaction between surface ligand and counterions. Constructing nanoparticles assembly with predictable and controllable structures and atom-accuracy ordering stands as one of the greatest challenge in materials science. Two types of Au nanoparticles: $[\text{Au}_{21}(\text{SR})_{12}(\text{PCP})_2]^+[\text{AgCl}_2]^-$ and $[\text{Au}_{21}(\text{SR})_{12}(\text{PCP})_2]^+[\text{Cl}]^-$ are studied in the present work. The PCP ligand and anions (AgCl_2^- or Cl^-) on nanoparticle surface exhibit strong π - π and anion- π interactions which can be used as surface hooks to direct the formation of 1-D assembly. The assembled structure is carefully studied by single crystal XRD. The authors also studied the electrical transport properties of the two types of nanoparticles assemblies and found that the one with Cl^- anion shows higher conductivity. It is a very well-written manuscript and a good addition to the materials design methods with complex structures, which should be disseminated to the broad audience. I have a few of minor comments:

[1] The authors should provide the monodispersity information of $[\text{Au}_{21}(\text{SR})_{12}(\text{PCP})_2]^+[\text{AgCl}_2]^-$ and $[\text{Au}_{21}(\text{SR})_{12}(\text{PCP})_2]^+[\text{Cl}]^-$ they synthesized. MALDI-MS data should be added. More details should be included, for example, any size selection process is used?

Response: Thanks for the comments. The MALDI-MS data of the new $[\text{Au}_{21}(\text{SR})_{12}(\text{PCP})_2]^+[\text{Cl}]^-$ has been added (Figure S1). No size selection process was used.

Revision: Newly added Figure S1.

Figure S1. MALDI-MS of the $[\text{Au}_{21}(\text{SR})_{12}(\text{P-C-P})_2]^+[\text{Cl}]^-$. An intense peak at ~ 6285 Da is assigned to the intact $[\text{Au}_{21}(\text{SR})_{12}(\text{P-C-P})_2]^+$ (theoretical value: 6288.25 Da). Note: The lower mass peaks are fragments caused by the 337 nm laser in MALDI-MS analysis. The intensities of fragments show a laser power dependence and the spacing of fragments and intact nanocluster is $(\text{AuSR})_n$ (n : 1-4). The measurement was done in positive mode.

[2] For electrical transport measurement, conductivity deviation and error bar for each assembly in different testing should be provided and commented.

Response: A box-and-whisker plot for the conductivity data set has been added to SI (Figure S7), showing the median, 25 and 75% quartiles, and the range. There is a large range in the measured conductivities, especially for the crystals with AgCl_2 counter-ion, but a clear and significant difference can still be seen between the two sets. Given the fragility and very resistive nature of the samples, the range can be attributed to variation in the quality of devices.

Revision: Newly added Figure S7:

[3] It is interesting to see the two orders of magnitude conductivity difference between two types of assemblies with different anions. The authors claim that this conductivity difference is not related to a small change of interparticle distance between two assemblies (0.41 \AA), as the previous results of the conductivity measured from Au NP films in ref 44-46 only exhibit 1 order of magnitude change for every 1 \AA increase of interparticle distance. But considering both conductivity values in the present work are very low (10^{-6} to 10^{-8} S/m, can be treated as insulator), this conclusion may be questionable. The 1 order of magnitude conductivity change with 1 \AA interparticle distance increase in ref 44-46 is observed in the

range of 10⁻¹ to 10⁻³ S/m where 1 order of value difference is very dramatic. Comparing to this, the conductivity change in the present work is not as significant as claimed, and it is not convincing to rule out the interparticle distance effect. The authors should comment more on this point.

Response: Thanks for the comments. In Ref 46, the result of “1 order of magnitude conductivity change with 1 Å interparticle distance increase” was observed in a *wide range* from 0 to 10⁻¹⁰ S/m (see the copied Figure 5 from ref 46), which indeed covers our samples’ conductivity range. We have further clarified our discussion as suggested by the reviewer.

Figure 5. Plot of 70 (○), 30 (●), and -60 °C (▼) conductivity vs number of carbons in the alkanethiolate chains of Au₃₀₉(C_n)₉₂ MPCs (eq 7). The inset is a schematic describing the interdigitation of monolayer chains in solid-state MPC films.

Revision: comments added on page 9,

“The significantly different conductivity (by two orders of magnitude) with such a small change of interparticle distance (0.41 Å) contrasts with previous experimental results of the conductivity measured from Au NP films^[43-46] (~1 order of magnitude for every 1 Å increase of interparticle distance). This suggests that besides the well-known interparticle distance factor, other effects could also be responsible for the large difference in electron transport in these NP crystals. In previous work, this was however unable to be further analyzed due to the lack of atomic-level characterization.”

REVIEWERS' COMMENTS:

Reviewer #1 (Remarks to the Author):

The revised manuscript is much improved. I believe it is suitable for publication in Nature Communications.

Reviewer #3 (Remarks to the Author):

The revised manuscript has well addressed all my questions/comments and those from other reviewers. I strongly recommend its publication as is in Nature Communications.